# In-Flight Attitude Control of a Quadruped using Deep Reinforcement Learning

**Tarek El-Agroudi**\*    **Finn Gross Maurer**\*    **Jørgen Anker Olsen**    **Kostas Alexis**

Norwegian University of Science and Technology

O. S. Bragstads Plass 2D, 7034 Trondheim

{tareke, finngm, jorgen.a.olsen, konstantinos.alexis}@ntnu.no

Paper video    GitHub IO page    Hardware design    Code

**Abstract:** We present the development and real world demonstration of an in-flight attitude control law for a small low-cost quadruped with a five-bar-linkage leg design using only its legs as reaction masses. The control law is trained using deep reinforcement learning (DRL) and specifically through Proximal Policy Optimization (PPO) in the NVIDIA Omniverse Isaac Sim simulator with a GPU-accelerated DRL pipeline. To demonstrate the policy, a small quadruped is designed, constructed, and evaluated both on a rotating pole test setup and in free fall. During a free fall of $0.7$ seconds, the quadruped follows commanded attitude steps of $45$ degrees in all principal axes and achieves an average base angular velocity of $110$ degrees per second during large attitude reference steps.

**Keywords:** Deep Reinforcement Learning, Legged Robotics

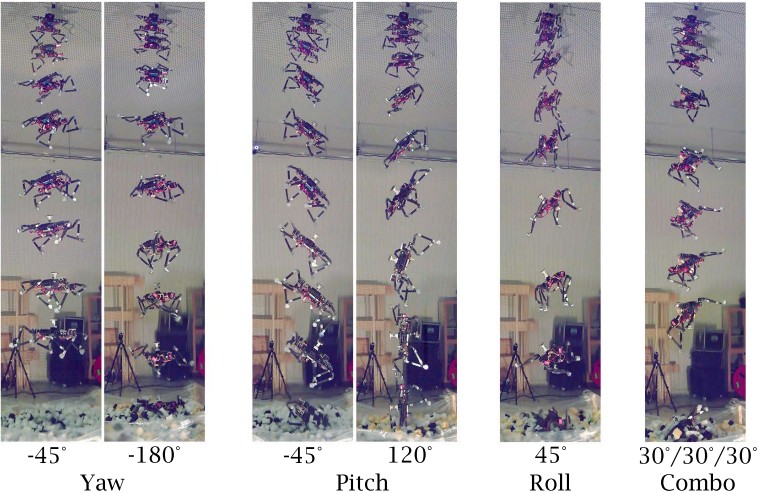

|  -45°  |  -180°  |  -45°  |  120°  |  45°  |  30°/30°/30°  |
| :---: | :---: | :---: | :---: | :---: | :---: |
|  Yaw  |  |  Pitch  |  |  Roll  |  Combo  |

Figure 1: Camera frames for selected free fall experiments, with commanded attitude setpoints.

## 1 Introduction

The exploration of extraterrestrial environments with robots is a daunting endeavor. On one hand, the impossibility of fast human intervention requires unprecedented robustness and autonomy while, on the other, the complexity and uncertainty of extraterrestrial surfaces requires dexterous locomotion. One particularly complex class of environments are martian and lunar lava tubes, the exploration of which might provide valuable geological insights and the potential for early human bases [1]. Wheeled rovers, of which several have been deployed on Mars and on the Moon [2], are inadequate

---

\*Indicates equal contribution.

8th Conference on Robot Learning (CoRL 2024), Munich, Germany.

to explore these terrains due to the need for significant vertical motion for lava tube exploration. Helicopters, such as Ingenuity [3], have also been demonstrated on Mars but have limited payload-carrying capacity. With the recent advances in quadrupedal robots (quadrupeds) such as the Boston Dynamics Spot [4], the ANYbotics ANYmal [5] and the Unitree Go 1 [6], a proposed solution for the robotic exploration of lava tubes has been to use specially designed quadrupeds that can exploit the lower gravity experienced on Mars or on the Moon by jumping and experiencing long phases of "flight" to seamlessly traverse complex terrain [7].

In order to cancel out any accumulated angular velocity during a prolonged flight-phase and to enable landing on inclined surfaces, an in-flight attitude[1] control law is a prerequisite for such a robotic platform. A proposed solution for the actuation of such capabilities is for the quadruped to use its own legs as reaction masses in maneuvers that resemble the self-righting motions of a falling cat. This has certain advantages compared to other orientation approaches, such as the reaction wheels exploited in satellites [8]. Primarily, the considered approach does not require extra hardware and thus keeps the overall system weight smaller, which has significant mission payload implications.

With the recent advances in simulation technologies and parallelization hardware, many complex control tasks have been solved with DRL-based approaches. DRL allows for the exploitation of the equivalent of many hours of real-time experience in just a few minutes [9], while also being able to generate lightweight control policies implementable on resource-constrained hardware. This motivates the use of DRL for synthesizing an in-flight attitude control law for a jumping quadruped and the implementation of such a policy on a quadruped in real life.

Driven by the above, the main contribution of this work is the development of a DRL-based attitude-control policy for a quadruped with a five-bar-linkage leg design and lateral leg movement capabilities, and the demonstration of this policy on a custom-designed prototype in all degrees of freedom (DOFs). The proposed test setups include maneuvering on a rotating pole and free fall experiments (a subset of the results is depicted in Figure 1) with the system following commanded attitude setpoints. The considered testing setups are inexpensive and easy to replicate. Furthermore, the developed quadruped design features low-cost hardware with of-the-shelf actuators and electronics, demonstrating the potential for robust DRL without exact knowledge of factors like actuator dynamics. Key insights that transfer to related robotic hardware are the treatment of low-cost RC servo motors through a reference model and motor model, and the explicit simulation of the closed kinematic chain of the robot's legs. We further show how the same reward function can be used to learn a policy for both fast and slow dynamics. To the authors' best knowledge, this work is the first real-world demonstration of an in-flight attitude-control feedback policy for a quadruped in 3D.

This paper is structured as follows: Related work is outlined in Section 2, while Section 3 formally defines the control problem to be solved using DRL. An overview of the robotic research platform and experimental setups is provided in Section 4, and the proposed approach for training and deploying the DRL-based policy is outlined in Section 5. Section 6 evaluates the experimental results, and Section 7 outlines central conclusions, limitations and future work.

## 2 Related Work

This work builds upon a range of previous contributions. The work in Rudin et al. [10] presented SpaceBok, a quadruped specifically designed for the exploration of lunar surfaces, and demonstrated a DRL-based policy in 2D on a test bed designed for micro-gravity experiments [11]. Kurtz et al. [12] demonstrates self-righting on the quadruped Mini Cheetah by restricting attitude errors to pitch and utilizing a combination of trajectory optimization and supervised learning techniques. The work on the tripedal robot SpaceHopper [13] has trained and demonstrated a DRL-based policy for attitude control in 3D on a parabolic flight and a specially designed gimbal setup. Furthermore, Olsen and Alexis [14] presented OLYMPUS, a quadruped with a five-bar-linkage leg design tailored for extraterrestrial exploration. The modeling and control of closed kinematic chains (CKC), such as the N-bar-linkage legs of OLYMPUS and SpaceBok, has been extensively researched [15, 16],

---

[1]We use attitude to denote the 3D orientation of the quadruped.

along with free-floating dynamical systems with manipulator structures [17]. Kamidi et al. [15] explicitly simulates the CKC and SpaceBok establishes an open-loop equivalent. For the synthesis of the policy using DRL, this work builds heavily upon [9], which demonstrated the learning of high-performance locomotion policies for quadrupeds in mere minutes by exploiting GPU parallelization. The main technique used for sim2real transfer of the learned policy for SpaceBok and SpaceHopper is domain randomization, which has been evaluated against a range of sim2real methods in [18].

## 3    Problem Formulation

With reference to Figure 2, the considered control problem can be formally defined as follows: Given the orientation quaternion[2] $q_b^w$ of the quadruped's body frame $\mathcal{B}$ in an inertial world frame $\mathcal{W}$ and a target orientation $\mathcal{T}$ expressed in $\mathcal{W}$ through $q_t^w$, control the quadruped towards $q_b^w = q_t^w$. The latter is equivalent to controlling the angle of the axis-angle representation of the quaternion error $q_b^t = (\mathbf{q}_t^w)^* \otimes \mathbf{q}_b^w$ to zero, denoted $\angle q_b^t \to 0$, which motivates the reward function presented in Section 5. The inputs made available to the control law are the the quaternion error $q_b^t$, its angular velocity $\omega_b^t$ and the positions and velocities of the 12 actuators, $\theta_m$ and $\dot{\theta}_m$, forming the observation vector $o = [q_b^t, \omega_b^t, \theta_m, \dot{\theta}_m]^T \in \mathbb{R}^{31}$. Actuation is restricted to the 12 actuators of the quadruped, and the output of the control law is thus a vector of motor position references $\theta_r \in \mathbb{R}^{12}$ sent to the motors, which in turn apply a torque $\tau_m \in \mathbb{R}^{12}$ to the actuated joints. We note that the target application of canceling angular momentum from a jump is a velocity tracking problem rather than a regulation problem. However, if position tracking is demonstrated, velocity tracking can be achieved by the introduction of a moving position target.

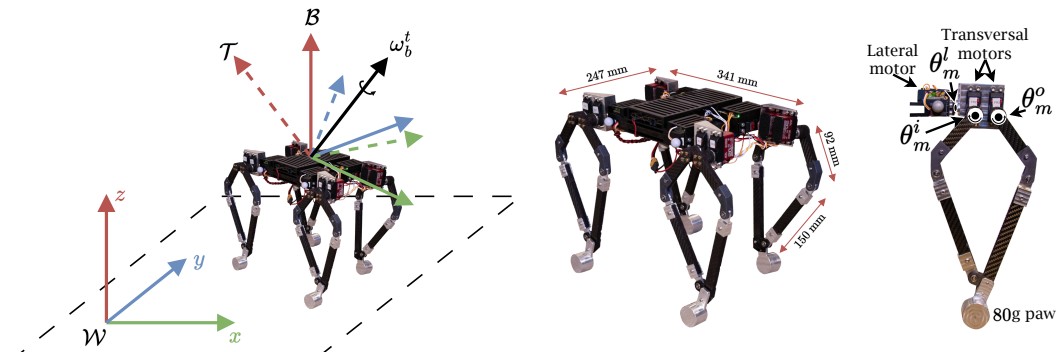

Figure 2: World, body and target frames for the attitude control problem. The black arrow indicates the angular velocity vector.

Figure 3: Eurepus dimensions and labeled leg. $\theta_m^i$, $\theta_m^o$ and $\theta_m^l$ are the inner transversal, outer transversal and lateral motor angles.

## 4    System Overview

The proposed research platform, called Eurepus, consists of four legs with a five-bar-linkage design, each consisting of a lateral motor and two transversal motors. The base is constructed with interior rails to allow the quadruped to be mounted on a forked rotating pole for single DOF attitude-control tests. The quadruped has a weight of 2.5 kg and the dimensions presented in Figure 3. Key design considerations include aluminum joints and carbon-foam links to ensure low weight and robustness. 80 gram aluminum paw masses equip the quadruped with the necessary inertia for using the legs as reaction masses, and constitute about an eighth of the system weight. The IB53BHP servo from AGF-RC is used for all three motors on a leg. It is a fast (180 RPM), light-weight (70 g), high-torque (2Nm) and low-cost servo with potentiometer position feedback. The motors are driven by a PWM driver and powered by a 2 cell LiPo battery. Position feedback is converted from analogue potentiometer readings via 12 ADS1115 ADCs. The Khadas Vim 3 (A311D) SBC is the central

---

[2]We use the Hamilton convention of the quaternion that defines a transformation from a frame $\mathcal{B}$ to frame $\mathcal{W}$ according to $x^w = q_b^w \otimes x^b \otimes (q_b^w)^*$, where $\otimes$ and $*$ denote the quaternion product and conjugate respectively.

control unit of the quadruped. To test the system at lower joint velocities and over extended periods of time, a stiff forked aluminum pole is mounted to an open ball bearing, such that the quadruped can be slid onto it in all principal axes, illustrated in Figure 4. The pole fixes the center of rotation and restricts attitude errors to 2D. We therefore conduct 3D free fall experiments, shown in Figure 5, by raising the quadruped into a mount designed to lock it into a known initial configuration. A magnetic switch is installed to remotely release the quadruped into a foam pit. The resulting configuration allows for a free fall duration of 0.7 seconds. Attitude feedback is provided through a motion capture (MoCap) system for the free fall tests and through an encoder for the pole setup.

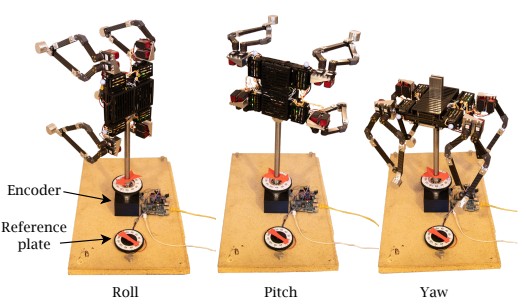

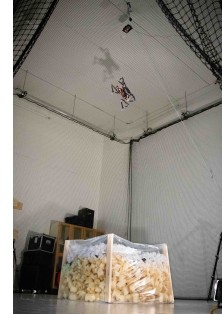

Figure 4: Rotating pole experimental setup.    Figure 5: Quadruped in a free fall pitch test.

## 5    Proposed Approach

We create a simulated environment of the Eurepus quadruped in NVIDIA Omniverse Isaac Sim and simulate 4096 environments in parallel to collect action-state-reward trajectories. The modified PPO implementation from [9] is leveraged to learn a control policy from the collected trajectories during simulation. Two policies are learned, for slow and fast motor velocities respectively. Implementation wise, the work builds on the OmniIsaacGymEnvs repository [19] for GPU-accelerated DRL.

### 5.1    The Reward Function

The reward function is defined as a weighted sum of eight reward terms, $R_{\text{total}} = \sum_{i=1}^{8} C_i \cdot R_i$, where the terms $R_i$ and scaling factors $C_i$ are given in Table 1. The terms are split into three mostly positive reward terms and five negative penalty terms. $R_1$ rewards closeness to the target orientation. To encourage small orientation errors, it is constructed as an exponential kernel with temperature $T_{\text{orient}}$ around the absolute value of the error angle $\angle q_b^t$, where the latter was defined in Section 3. $R_2$ provides an extra discrete reward whenever the quadruped is within a threshold $\epsilon_{\text{orient}}$ radians of the target, where $\epsilon_{\text{orient}} > 0$ is tuned according to the required accuracy. Finally $R_3$ acts as a sparse reward provided only when the quadruped terminates due to a success criteria of having been within $\epsilon_{\text{orient}}$ degrees of the target for more than half a second consecutively. Its design renders it negative when the orientation error is larger than 90 degrees and positive when it is less.

| $i$ | **Reward/Penalty** | $R_i$ | $C_i$ |
|---|---|---|---|
| 1 | Orientation Reward | $\exp\left(-\lvert\angle(\mathbf{q}_b^t)\rvert/T_{\text{orient}}\right)$ | 2 |
| 2 | Inside Threshold Reward | $\begin{cases} 1 & \lvert\angle(\mathbf{q}_b^t)\rvert < \epsilon_{\text{orient}} \\ 0 & \text{otherwise.} \end{cases}$ | 1 |
| 3 | Terminal Reward | $\frac{\pi}{2} - \lvert\angle(\mathbf{q}_b^t)\rvert$ | 100 |
| 4 | Self-Collision Clamping Penalty | $-\lVert\boldsymbol{\theta}_{\text{interpol}} - \boldsymbol{\theta}_r\rVert^2$ | $1 \times 10^{-6}$ |
| 5 | Torque Saturation Penalty | $-\lVert K_p(\boldsymbol{\theta}_r - \boldsymbol{\theta}_m) - K_d\dot{\boldsymbol{\theta}}_m - \boldsymbol{\tau}_{\text{max}}\rVert^2$ | $1 \times 10^{-2}$ |
| 6 | Velocity Penalty | $-\lVert\dot{\boldsymbol{\theta}}_m\rVert^2$ | $2 \times 10^{-5}$ |
| 7 | Torque Regularization Penalty | $-\lVert\boldsymbol{\tau}_m(t) - \boldsymbol{\tau}_m(t-1)\rVert^2$ | $3.5 \times 10^{-2}$ |
| 8 | Change Direction Penalty | $\begin{cases} -1 & \dot{\boldsymbol{\theta}}_m(t) \cdot \dot{\boldsymbol{\theta}}_m(t-1) < \epsilon_{\text{noise}} \\ 0 & \text{otherwise.} \end{cases}$ | $2 \times 10^{-3}$ |

Table 1: Reward function terms and their scaling factors. $\angle$ denotes the angle in radians.

With reference to Section 5.3, $R_4$ punishes the squared norm of the difference between interpolated motor targets, $\boldsymbol{\theta}_{\text{interpol}}$, and the motor targets after being clamped to prevent self-collisions, $\boldsymbol{\theta}_r$. Similarly, $R_5$ punishes the excess torque that is clipped away during saturation at maximum torques $\boldsymbol{\tau}_{\text{max}}$, where $K_p$ and $K_d$ are the parameters of the motor's interior PD controller. Both terms punish the setting of targets that are clipped or clamped in subsequent steps, and are inspired by [10]. The final reward terms $R_6$, $R_7$ and $R_8$ are regularization penalties and serve the purpose of enforcing smooth policies that minimize actuator effort, to reduce motor wear and tear. $R_6$ punishes the squared norm of the motor velocity vector and $R_7$ punishes the squared norm of the difference between consecutive applied torques, while both terms serve the purpose of minimizing wear and tear. Finally, $R_8$ is a discrete penalty given anytime the motors change direction, to enforce smooth maneuvers, where $\epsilon_{\text{noise}} > 0$ is a threshold included to exclude numerical noise.

A few important practical considerations were necessary for successful training. Since the loss function of PPO considers only the total reward $R_{\text{total}}$, the reward weights $C_i$ are chosen such that the reward terms are comparable in order of magnitude. Furthermore, implementing the orientation rewards as positive rewards instead of penalties was essential to avoid the local optimum of self-termination. The most important reward is the orientation reward $R_1$, and it alone suffices to achieve a working policy with PPO, but increased performance is experienced with the addition of the other terms. While self-collisions of a leg are prevented by explicit clamping, as will be outlined in Section 5.3, no explicit collision filter is implemented to avoid collisions between separate legs. Instead, the quadruped is reset upon the occurrence of collisions such that the accumulated reward after the trajectory is smaller (total reward is in general positive), thus implicitly incentivising collision-free maneuvers. Another approach that was explored was to explicitly punish collisions through negative penalties, but this led to conservative policies. The developed policy can be combined with an explicit collision filter during deployment on hardware. Finally, when training a policy for lower motor velocities, we use the exact same reward function as for the unconstrained velocity problem. We find that reducing the inference frequency and extending the time horizon parameter of PPO with the same ratio as the ratio of maximum motor velocities results in successful learning. This likely stems from the fact that a trajectory will contain similar degrees of exploration.

## 5.2   Simulation and sim2real

A series of steps were necessary to achieve a stable and realistic simulation in Isaac Sim. Isaac Sim simulates systems of rigid bodies through articulations, a reduced-coordinate representation which is fully determined by a root body and joint angles between all bodies. This mechanism does not support loops, which required separately modeling the ankle joint as a constraint. In Isaac Sim, this is achieved by excluding the joint from articulation. This constraint and the significant mass differences between the links of the quadruped render the dynamics stiff. To ensure stability of the resulting simulation model, the most important consideration was the need for a low simulation time step $\Delta t = 1/480s$, which is eight times smaller than the simulation step in [9].

A prerequisite for the successful sim2real transfer of the policy was an actuator model. The actuators are modeled by treating the internal motor dynamics as a static torque-speed mapping, $F_{\text{ts}}(\dot{\theta})$, where $\theta$ is the motor joint angle and $F_{\text{ts}}$ is a linear curve provided by the manufacturer. This simplification is based on the intuition that electrical dynamics are much faster than the mechanical dynamics $\dot{\theta}$. The parameters $K_p$ and $K_d$ of the actuator's internal PD controller were unknown and hence identified from step response data. The resulting simplified motor model is given by:

$$\tau = \text{sat}(K_p(\theta_r - \theta) - K_d\dot{\theta}, \tau_{\text{max}}), \quad \text{where} \quad \tau_{\text{max}} = F_{ts}(\dot{\theta}). \tag{1}$$

Table 2 presents the sim2real techniques employed during training in preparation for deployment on the real system, where $\mathcal{U}$ and $\mathcal{N}$ indicate sampling from uniform and normal distributions. The significant randomization of the controller parameters stems from the simplifications used in (1) and uncertainty in the system identification procedure. The other randomization parameters originate from the inaccurate potentiometer feedback from the low-cost actuators. Factors that were less uncertain, such as the system mass and inertia, are not randomized.

| Entity | Type of sim2real | Amount | Applied on |
|---|---|---|---|
| $K_p, K_d$ | Domain randomization | $\pm(20-50)\%$ | Reset |
| Motor positions measurement | Measurement bias | $\mathcal{U}(-2, 2°)$ | Reset |
| Motor positions measurement | Additive noise | $\mathcal{N}(0, 5°)$ | Measurement time step |
| Motor velocities measurement | Additive noise | $\mathcal{N}(0, 20°)$ | Measurement time step |

Table 2: Sim2Real variables

## 5.3 Policy Architecture and Deployment

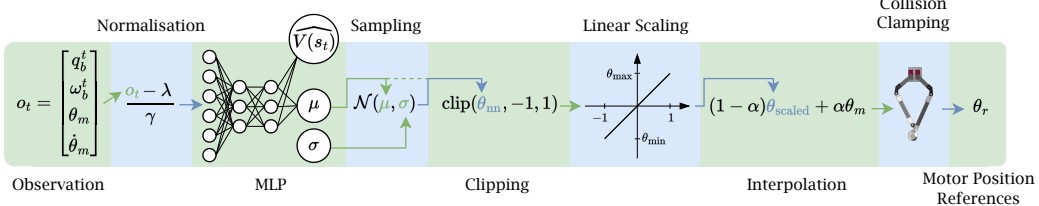

Figure 6: Architecture of the attitude-control policy, from observation $\boldsymbol{o}_t$ to motor references $\boldsymbol{\theta}_r$.

Figure 6 illustrates the policy architecture. The same network is used for both the actor and critic networks of PPO. The observation $\boldsymbol{o}_t$ is normalized by a running mean $\lambda$ and variance $\gamma$ which are updated during training, before being fed through a three layer feed-forward neural network with layer width $128$, $64$ and $64$, and exponential linear unit (ELU) activation. The network output consists of the value function estimate $\widehat{V(\boldsymbol{s}_t)}$ to form the critic, and the mean $\boldsymbol{\mu}$ and standard deviation $\boldsymbol{\sigma}$ of a Gaussian distribution to form the actor. The distribution serves the purpose of ensuring exploration during training. During deployment, the standard deviation is ignored, and the network output for the mean, denoted $\boldsymbol{\theta}_{nn}$, is used directly in the subsequent steps of the policy. Before being applied as joint targets to the motors, $\boldsymbol{\theta}_{nn}$ is clipped to the range $[-1, 1]$ and mapped to joint-specific motor limits $[\boldsymbol{\theta}_{\min}, \boldsymbol{\theta}_{\max}]$ to form scaled targets $\boldsymbol{\theta}_{\text{scaled}}$. These are then sent through a linear interpolation (2) with an exponential kernel interpolation factor $\alpha$ with temperature $T_{\text{interpol}}$, where $\boldsymbol{\theta}_m$ are the measured joint positions, in order to reduce control efforts around the target orientation.

$$\boldsymbol{\theta}_{\text{interpol}} = (1-\alpha)\boldsymbol{\theta}_{\text{scaled}} + \alpha\boldsymbol{\theta}_m, \quad \text{where } \alpha = \exp\left(-\frac{(\angle(\mathbf{q}_b^t))^2}{T_{\text{interpol}}}\right). \tag{2}$$

Finally, the interpolated joint targets $\boldsymbol{\theta}_{\text{interpol}}$ are clamped to avoid self collisions of a leg. This is implemented as a condition based on the sum $\theta_{\text{sum}} \doteq \theta_{\text{i}} + \theta_{\text{o}}$ of the inner and outer transversal motors angles $\theta_{\text{i}}$ and $\theta_{\text{o}}$, and an acceptable range for this sum $[\theta_{\text{min sum}}, \theta_{\text{max sum}}]$, according to the following conditions, where $-=$ and $+=$ are the decrement and increment operators:

| Condition | Clamping Operation | Condition | Clamping Operation |
|---|---|---|---|
| $\theta_{\text{sum}} > \text{threshold}$ | $\theta_{\text{i}} \mathrel{-}= \frac{(\theta_{\text{max sum}} - \theta_{\text{sum}})}{2}$ $\theta_{\text{o}} \mathrel{-}= \frac{(\theta_{\text{max sum}} - \theta_{\text{sum}})}{2}$ | $\theta_{\text{sum}} < \text{threshold}$ | $\theta_{\text{i}} \mathrel{+}= \frac{(\theta_{\text{min sum}} - \theta_{\text{sum}})}{2}$ $\theta_{\text{o}} \mathrel{+}= \frac{(\theta_{\text{min sum}} - \theta_{\text{sum}})}{2}$ |

If a policy is trained or deployed at reduced motor velocities, a smooth commanded target sequence $\boldsymbol{\theta}_c$ is achieved by sending the clamped and interpolated motor targets $\boldsymbol{\theta}_r$ through the saturated second order reference model given by (3). $\nu_c$ denotes the commanded velocity, and the absolute damping factor $\zeta$ and natural frequency $\omega_n$ are filter tuning parameters. A second order reference model is advantageous as it generates smooth velocities in addition to smooth position signals.

$$\dot{\theta}_c = \text{sat}(\nu_c), \quad \text{where} \quad \text{sat}(x) \doteq \begin{cases} \text{sgn}(x)x^{\max} & \text{if } |x| \geq x^{\max} \\ x & \text{else} \end{cases} \tag{3}$$
$$\dot{\nu}_c + 2\zeta\omega_n\nu_c + \omega_n^2\theta_c = \omega_n^2\theta_r$$

# 6 Evaluation Studies

To evaluate the performance of the learend policy on Eurepus, tests were conducted both per DOF on a rotating pole and during free fall 3D maneuvering. A comparison with other control approaches and specifically with Model Predictive Control is presented in the supplementary materials.

## 6.1 Rotating Pole

We evaluate a 3D policy trained with joint velocities below 300°/s through (3) against a time varying reference on the rotating pole setup. Step responses for roll, pitch and yaw are given in Figure 7. Figure 8 presents an ablation study of the paw masses. The policy successfully follows the reference for all DOFs, validating the test setup. The spike in yaw indicates that the policy has learned to sometimes take the longest path to a reference, and the degradation in the pitch DOF after 60 seconds originates from a leg collision and the corresponding accumulated angular velocity.

A key finding is that the 3D policy trained in a pole-less environment and with no variations of paw masses is robust to the effects of the pole as well as variations of the paw mass. This indicates that the learned feedback policy has not overfitted to the system dynamics and justifies the simple feed-forward network structure as opposed to an adaptive approach through for instance recurrence in the policy. A potential drawback of the policy is excessive control effort and oscillations at the reference. Possible reasons for this are noise in joint measurements, which directly transmits to the actuators through the linear interpolation (2), inadequate motor model (1) leading to excessive actuation or slower dynamics on the real system, and potential collisions of the legs with the pole.

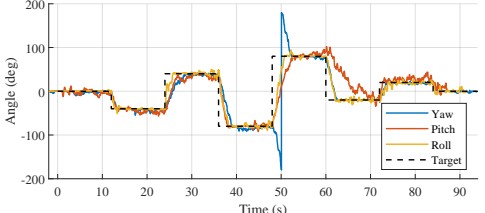

Figure 7: 3D policy response to references in yaw, pitch and roll on the real system.

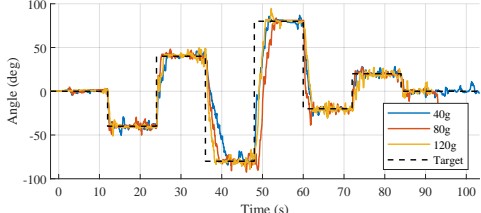

Figure 8: 3D policy roll response for different paw masses. Policy trained at 120 g.

## 6.2 Free Fall

We demonstrate attitude control of the quadruped through a range of free-fall experiments from a height of about 3.5 meters and landing into a foam pit. Figure 9 displays the simulated and real response of the error quaternion decomposed into Euler angles (ZYX convention) for 45 degrees setpoint in roll, pitch and yaw. Each experiment is repeated thrice and the plot indicates the mean and variance of the response to demonstrate repeatability. The target orientation is reached in less than 0.4 seconds for yaw and pitch, and 0.6 seconds for roll. Next, Figure 10a demonstrates non-principal axis attitude control by applying a simultaneous 30 degrees reference in roll, pitch and yaw. Finally, we show that the policy is capable of controlling 80 degrees of orientation during the free fall time of 0.7 seconds by applying large reference steps in pitch and yaw, as seen in Figures 10b and 10c. Extrapolating the results yields an estimated acheivable average attitude velocity of 110 degrees per second. Frames from a recording of selected experiments are provided in Figure 1.

Compared with the simulated trajectories, the real responses demonstrate robust sim2real transfer, but are simultaneously characterized by a slower transient and oscillations at the reference orientation. The main contributing factors to this are a) a time lag before a measurement is received from the MoCap system as well as b) a higher time constant in the real motor control loop than in the simulation, experimentally identified to be 50% higher for the real actuators. The former is estimated to be $\approx 0.03 - 0.05$ seconds, and such time lags can cause oscillations in feedback control systems dealing with fast dynamics as in Eurepus. The higher time constant in the actuators likely stems from the static approximation of the internal motor dynamics employed in the simplified motor model (1).

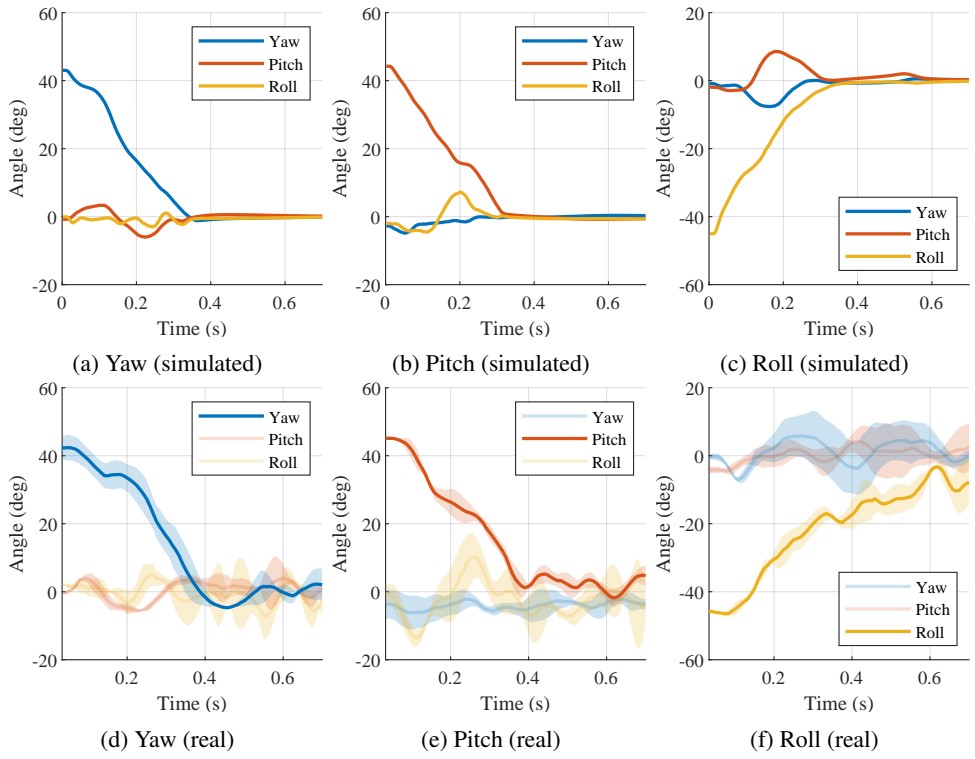

Figure 9: Euler angle errors for simulated and real robot for 45 degrees attitude steps.

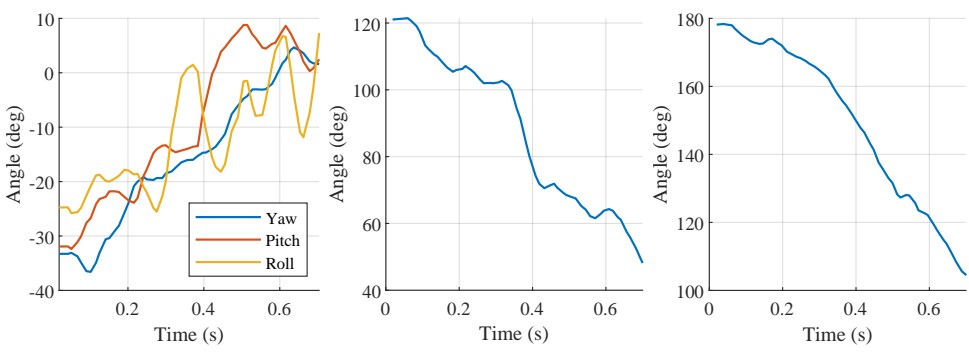

(a) 30 degrees roll, pitch and yaw  (b) 120 degrees pitch reference  (c) 180 degrees yaw reference

Figure 10: Selected results for free fall tests on the real robot. Plot (a) displays the decomposed error quaternion, and plots (b) and (c) display the absolute angle error.

## 7 Conclusions, Limitations and Future Work

We demonstrate fast attitude control on a custom-built low-cost quadruped with a five-bar-linkage leg design on a rotating pole test setup at lower speeds and in free fall. Tuning the horizon and inference frequency of PPO to replicate another problem with faster dynamics proved to be an effective way to relearn a problem for slower dynamics. Limiting assumptions of the work are a disturbance-free environment and the set-point-regulation problem formulation. Future work should consider more realistic extraterrestrial conditions and also address the velocity-tracking problem. Key limitations of the proposed approach are the simplified motor model and the possibility of leg collisions. This should be addressed through better actuator models and an explicit safety collision filter. Before operational system deployment, considerations should be made as to whether the learned attitude-control policy efficiently prepares the quadruped for landing.

**Acknowledgments**

This work was partially supported by the Research Council of Norway under Grant Agreement 338694. The authors extend their gratitude to Michalis Papadakis for providing an MPC baseline in the supplementary material. We would further like to thank the reviewers for their time and comments, and for improving the quality of the paper.

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

# Appendix
# Supplementary Material

## 1 Baseline Comparison with Model Predictive Control (MPC)

To motivate the use of DRL for solving the attitude-control problem, and to evaluate the learned policy's performance against classical control methods, we present the results for a model predictive control (MPC) approach to the attitude-control problem on Eurepus.

The implemented controller features a cascade structure, with an outer torso position controller to calculate the needed net torque, an inner torque-tracking controller to plan the leg motions of a single leg, and finally, a symmetry-based allocation strategy to map the leg motions to the remaining three legs in a collision-free manner. Furthermore, the controllers are encased in a finite state machine (FSM) which changes between a torque-applying stage and a predefined resetting stage when the legs have reached the endpoints of their range of motion. An important implication of the enforced symmetry in the leg manoeuvres is that only principal-axis rotations (yaw, pitch, roll) are possible. Hence, the control approach achieves non-principal-axis rotations by regulating one DOF at a time, in order of decreasing regulation error. This switching is also handled by the FSM. The described approach represents one possible avenue to design a MPC solution to the problem.

The controller is implemented with the simulation model of Eurepus and tuned for regulation performance. The performance of the controller against the DRL-based policy is presented in Figure 1 when given a 45-degree reference in yaw, pitch and roll. One central reason for the inferior performance as compared to that of the DRL-based policy is that the simplifications made in the controller restrict the space of available manoeuvres at any time. Such measures include in particular the symmetry restriction and the enforcing of an explicit retraction and resetting stage through the FSM.

While the disadvantage pointed out above is implementation-specific, it generalizes to classical optimal control approaches. While DRL directly optimizes the control task's objective, the optimal control approach often needs to split this objective into the sub-problems of planning and control (primarily for feasibility and computational tractability reasons), and the restrictions imposed by this can limit the range of behaviour significantly [1].

Several other factors put the MPC-based method at a disadvantage to the DRL-based policy. A well-known disadvantage of such receding horizon control methods is their computational cost which is typically high. Specifically, the computation time of the implemented MPC is about 13 ms, which is to be compared with the inference time of 0.06 ms for the DRL-based policy network. This additional processing time in the control loop has a detrimental impact both on the computational resources used and on the achievable control frequency during deployment. In particular, the control frequency of 80 Hz used for the DRL-based policy is not achievable with the MPC. This is because the latency in the other parts of the real-time control loop already comprises about 8 ms. Adding the computational time of the MPC results in a maximum achievable control frequency of 50 Hz. A reduction in control frequency is a disadvantage because of the particularly short gait during the free fall experiments and the requirement for extremely fast actuation in the target application.

---

[1]Y. Song, A. Romero, M. Müller, V. Koltun, and D. Scaramuzza. Reaching the limit in autonomous racing: Optimal control versus reinforcement learning. Science Robotics, 8(82), Sept. 2023. ISSN 2470-9476. doi:10.1126/scirobotics.adg1462. URL http://dx.doi.org/10.1126/scirobotics.adg1462.

Finally, tuning of the MPC is no trivial task, due to the cascade structure and number of parameters. While tuning is also non-trivial for the proposed DRL-based method, we find the required tuning efforts comparable, such that ease of tuning is not an advantage of the MPC-based approach.

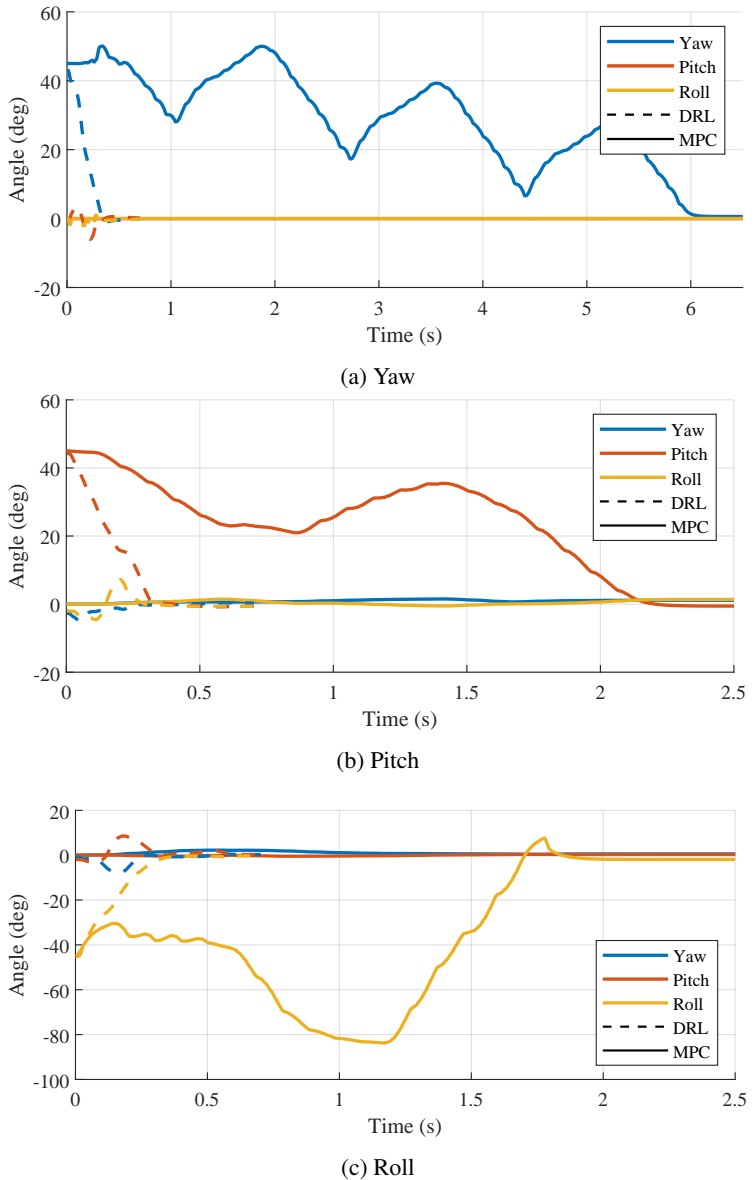

(a) Yaw

(b) Pitch

(c) Roll

Figure 1: 45-degree setpoint regulation performance for DRL-based policy (dashed line) and MPC-based controller (solid line) in simulation. The plot shows angle errors.

ii

## 2   Motor Model Ablation

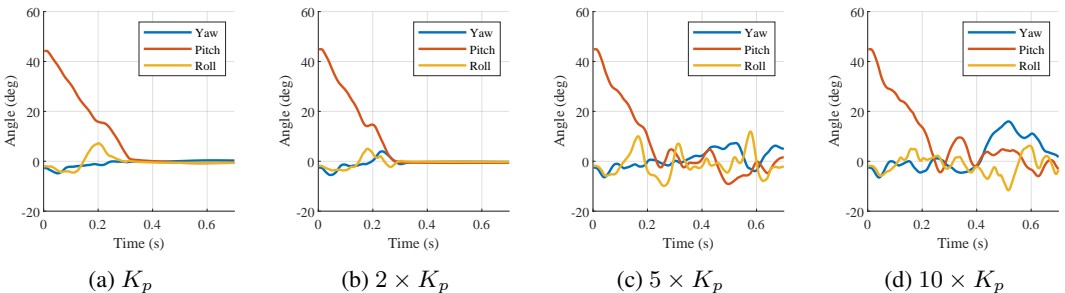

(a) $K_p$     (b) $2 \times K_p$     (c) $5 \times K_p$     (d) $10 \times K_p$

Figure 2: Response to a 45 degrees reference in pitch for different motor dynamics, where $K_p$ has been modified.

To assess the impact of the motor model on the performance of the policy, we repeat a 45-degree pitch experiment with four different configurations of the $K_p$ gain of the motor controller in simulation (Figure 2). $K_p$ is the nominal proportional controller gain and is the one employed during training. The ablation demonstrates degraded performance from changes to the gain indicating the importance of an accurate motor model. We further note the similarity between the degraded performance with the real experimental results of Figure 9d-9f in the paper, which may highlight the merit of further effort into motor modelling, through for instance an actuator network.

## 3   Angular Velocity Tracking

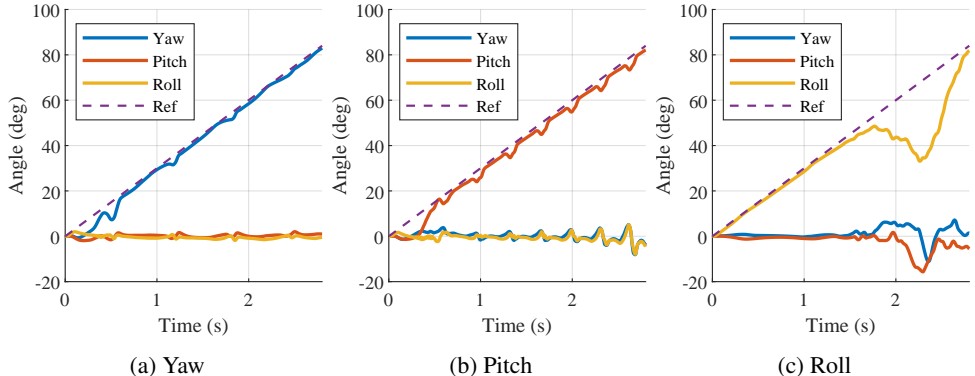

(a) Yaw        (b) Pitch        (c) Roll

Figure 3: 30 degrees per second velocity tracking performance of the attitude control policy in yaw, pitch and roll. The reference angle is indicated by a dashed line.

To assess the suitability of the developed policy to velocity tracking problems, we evaluate the velocity tracking performance to ramp references with a rate of 30 degrees per second in all principal axes (Figure 3). The velocity evaluated is motivated by the estimates of the angular velocity following a jump derived in related work[2]. The oscillation observed around the target is a result of the interpolation step of the policy. However, tracking is demonstrated in yaw and pitch. Inferior tracking is observed in roll, although a velocity of 30 degrees per second in roll is more than what can reasonably be expected from a normal jump according to [2]. This further clarifies the applicability of the learned policy to offset any initial angular velocity from a jump.

---

[2]A. Olsen and K. Alexis. Martian lava tube exploration using jumping legged robots: A concept study. In 74th International Astronautical Congress (IAC), Baku, Azerbaijan, October 2-6 2023. IAF.

## 4 Training Times

We present in Table 1 the training times for the two developed DRL-based policies. The GPU used is the NVIDIA GeForce RTX 3090, and the GPU memory usage was 4GB out of the available 24 GB RAM. The training time for the low-motor-velocity policy for the pole experiments is slower due to the inherently slower dynamics. Furthermore, training times are generally slower than similar related work using GPU-parallelisation[2]. This is mainly attributed to the need for extremely small simulation time steps due to stability issues of the Closed Kinematic Chain(CKC) simulation.

Table 1: Training time and epochs until convergence for trained policies

| Policy | Training time (hrs) | Number of Epochs |
|---|---|---|
| 3D pole (slow) | 7.0 | 800 |
| 3D free fall (fast) | 2.6 | 1500 |

## 5 Reduced Reward Function

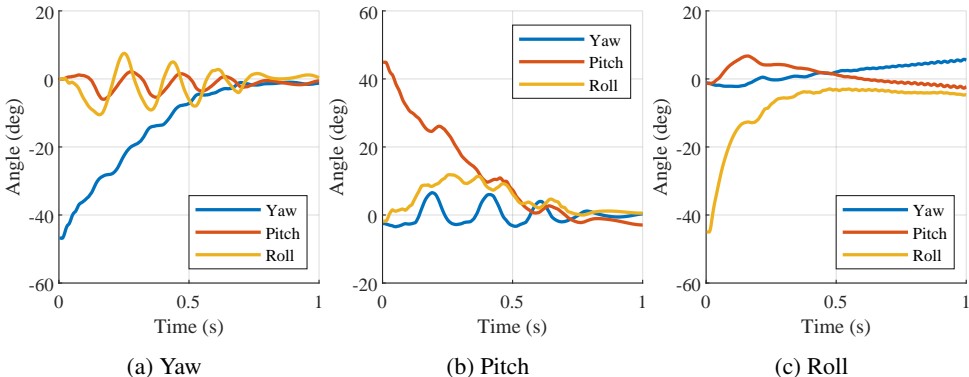

(a) Yaw          (b) Pitch          (c) Roll

Figure 4: 45-degree step responses where the policy is trained with only the three first rewards in Table 1 in the manuscript.

We train and evaluate the performance of a policy trained with a reduced reward function comprising only the first three reward terms, $R_{\text{total}} = C_1 R_1 + C_2 R_2 + C_3 R_3$. The five terms omitted correspond to the negative penalty terms of the original reward function. The scaling factors are not altered.

The results, in simulation, of 45-degree setpoint following in roll, pitch and yaw are provided in Figure 4. The results show that successful attitude control policies can be trained with the reduced reward function, which demonstrates that the main task objective is sufficiently represented by only three terms. An important observation is the oscillatory response when compared to the responses presented in the work. This is a natural consequence of the removal of the penalty terms which reduce actuator wear and tear (reward terms $R_5, R_6$ and $R_7$) and directly punish oscillatory motor action (reward term $R_8$).

## 6 Hardware Design

The proposed robotic platform is designed to be modular, and all parts can be 3D printed. CAD and STL files will be open-sourced, such that the robot can be replicated easily.

As is illustrated in Figure 6, the base of Eurepus consists of three main plates. The four legs slide onto the corners of a central base plate through the four lateral motor mounts. The electronics are

[2]N. Rudin, D. Hoeller, P. Reist, and M. Hutter. Learning to walk in minutes using massively parallel deep reinforcement learning, 2022.

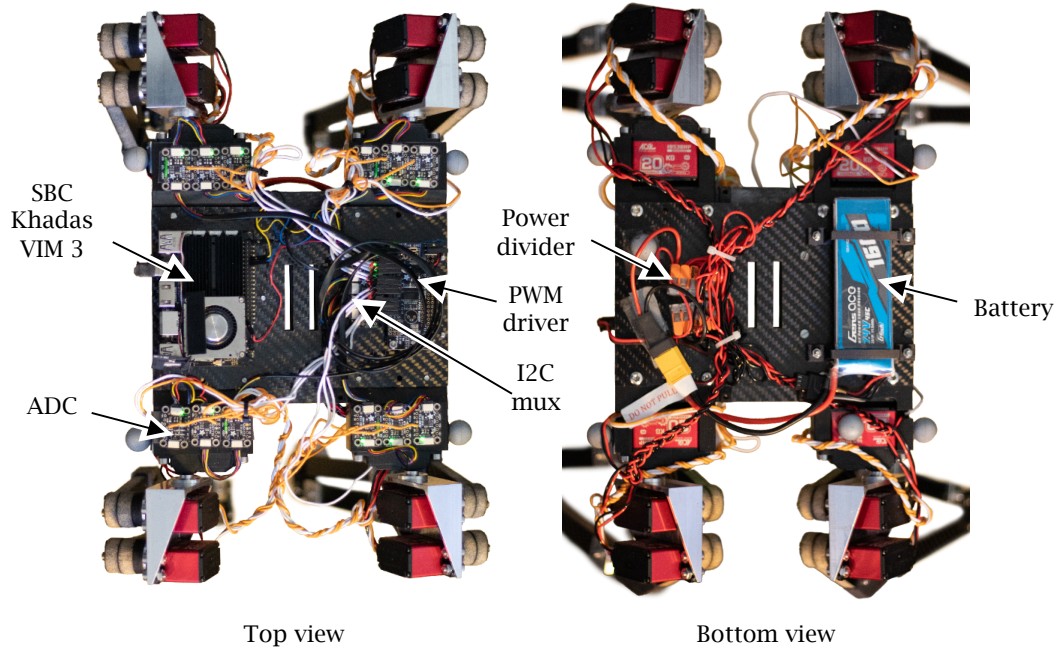

SBC Khadas VIM 3

ADC

Power divider

PWM driver

I2C mux

Battery

Top view

Bottom view

Figure 5: Picture of the electronics on Eurepus.

screwed to the upper and lower electronics plates and fit on top of the lateral motor mounts. The space between the base plate and the electronics plates is occupied by a configuration of rails. These, along with the parallel holes in the centre of each plate, are designed to allow a forked pole to be slid through the quadruped in all three principal axes. To protect the electronics on the upper plate and the ADCs mounted on top of the lateral motor mounts, a base cage and ADC cage are fitted over these. The entire configuration is fastened through three screws in each corner of the plates.

Each leg consists of a lateral motor and two transversal motors. The transversal motors fit snugly into a transversal motor mount that is mounted directly to the rotational axis of the lateral motor. The two thighs are mounted to the transversal motors with spacer plates to provide additional room when the legs traverse over the base. The leg itself is held together by parts designed to clamp to a thigh or shank on one end and connect in a ball-bearing joint on the other end. The paw clamp at the bottom of the leg contains additional fastening mechanisms for the paw mass.

Finally, we present an annotated picture of the electronics on Eurepus in Figure 5 to highlight how they fit in the mechanical design.

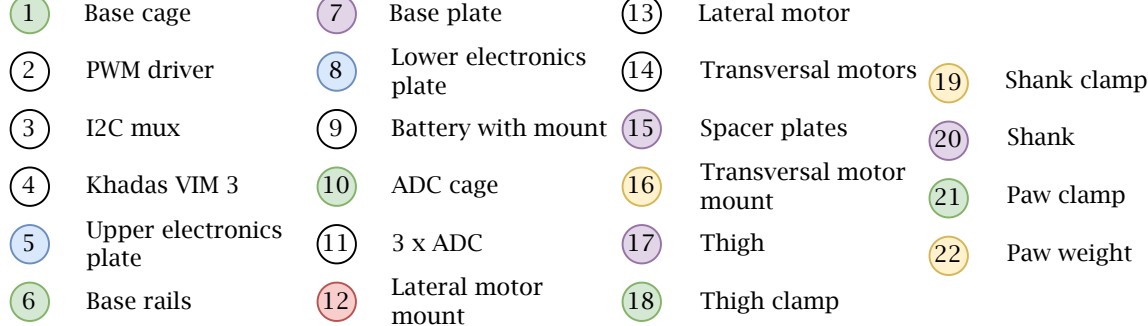

| | | | | | | | |
|---|---|---|---|---|---|---|---|
| ① Base cage | ⑦ Base plate | ⑬ Lateral motor | | | | | |
| ② PWM driver | ⑧ Lower electronics plate | ⑭ Transversal motors | ⑲ Shank clamp | | | | |
| ③ I2C mux | ⑨ Battery with mount | ⑮ Spacer plates | ⑳ Shank | | | | |
| ④ Khadas VIM 3 | ⑩ ADC cage | ⑯ Transversal motor mount | ㉑ Paw clamp | | | | |
| ⑤ Upper electronics plate | ⑪ 3 x ADC | ⑰ Thigh | ㉒ Paw weight | | | | |
| ⑥ Base rails | ⑫ Lateral motor mount | ⑱ Thigh clamp | | | | | |

Figure 6: Exploded view of the Eurepus quadruped design.

