# OpenReview forum: "In-Flight Attitude Control of a Quadruped using Deep Reinforcement Learning"
_robot-learning.org/CoRL/2024/Conference — CoRL 2024_

### Official Review · Reviewer_LWaG · 2024-07-08
**Review for submission 354**

**Originality:** 3
**Technical Quality:** 3
**Clarity Of Presentation:** 3
**Potential Impact:** 3
**Recommendation:** 4
**Confidence:** 4

**Review:**

This paper proposed a modified PPO approach for Quadruped robot control.

Strength:
The target task is interesting.
Real-world robot experiment was conducted.

Weakness:
The novelty of the proposed method based on PPO should be further explained.
The impact of the proposed method to the superior experimental results is not detailed.
Comparison with other RL baseline seems not provided.

More information related to the hard ware and RL setting should be added. More figures related to real-world task are welcome.
Why the original PPO in [9] can not be directly used in this task? The impact of the Modified PPO to the target task should be carefully analyzed.
It is suggested to discuss the potential of the proposed method to pure real-world task without relying on simulation and sim2real. Also the generalization ability of the proposed method when the real hardware and simulation are mismatch is worth discussing.
It is also suggested to add comparison with other RL baselines to further demonstrated the advantage of the proposed method.

**Quality Of The Limitations Section:**

2

**Questions For Rebuttal:**

More information related to the hard ware and RL setting should be added. More figures related to real-world task are welcome.
Why the original PPO in [9] can not be directly used in this task? The impact of the Modified PPO to the target task should be carefully analyzed.
It is suggested to discuss the potential of the proposed method to pure real-world task without relying on simulation and sim2real. Also the generalization ability of the proposed method when the real hardware and simulation are mismatch is worth discussing.
It is also suggested to add comparison with other RL baselines to further demonstrated the advantage of the proposed method.

**Robotics Focus:**

4

**Summary Of Paper:**

A modified PPO for quadruped robot in in-flight attitude control

**Summary Of Recommendation:**

More detials of real-world experiment and some comparsion with RL baselines could further improve this paper. It is also suggested to explain the proposed method as one unique approach for the target task rather than just a modified PPO.

---

### Official Review · Reviewer_82XZ · 2024-07-17

**Originality:** 3
**Technical Quality:** 3
**Clarity Of Presentation:** 4
**Potential Impact:** 3
**Recommendation:** 3
**Confidence:** 4

**Review:**

UPDATE: updated to weak accept after rebuttal.

While the results presented in this paper are a bit incremental, I find that the authors put significant effort in the experiment, and the paper provides a lot of insights besides the results. The paper currently has a few issues in results and presentation, which is why I'm giving a "weak reject" at this moment. However, I feel this paper could merit an acceptance after revision.

**Strengths:**

1. Detailed evaluation in simulation and real-world.

The authors presented detailed results for single-pole results (Section 6.1) and free-fall results (Section 6.2) with ablation studies and sim-to-real comparisons. These results clearly demonstrates the capability of the learned controller, which is greatly appreciated.

2. Clear and thorough presentation

I enjoyed the author's detail-oriented presentation of the proposed method, including the environment setup (last paragraph of Section 5.1), simulator configuration and sim-to-real (Section 5.2), policy representation (Section 5.3). These details are often critical in ensuring the successful deployment of a reinforcement learning policy. However, they are usually omitted from the paper, and can only be found in the Appendix, open-sourced code (or not found at all). In contrast, I feel the authors presented these important details in a thorough, coherent and easy-to-follow manner.

**Weaknesses:**

1. The result is incremental.

Given the current status of DeepRL for legged locomotion (esp. [10]), the result from this paper feels a bit incremental, and does not demonstrate a lot of novelty compared with existing works. Would the learned attitude controller lead to better pose control of the robot in difficult movements? Would the controller improves the balance of the robot? Extending the paper in some of these directions could improve the impact of the work.

2. Lack of comparison with existing work.

While the results presented are complete and thorough, the authors should compare their results with existing frameworks to better demonstrate their proposed methods (e.g. [10]). Would the method in [10] directly lead to good performance on the author's hardware, or would their be additional designs?

**Quality Of The Limitations Section:**

3

**Questions For Rebuttal:**

I would like the authors to answer the following questions in the rebuttal:
* In section 5.1 (Line 112), the authors mention that "two policies are learned" for slow and faster motor velocities. What is the motivation behind the two-policy design? Which policy is used for the evaluation results in Section 6? How do you determine which policy to use, and how do you handle the switch between policies?

* Is the result presented in section 6.1 in simulation or in the real world?

* How would the proposed method compared with the results in [10]?

**Robotics Focus:**

4

**Summary Of Paper:**

This paper proposes a method to use reinforcement learning (RL) for in-flight attitude control of a quadrupedal robot. The setup is discussed in detail, and both simulation and real-world experiment results are presented.

**Summary Of Recommendation:**

Despite the lack of ground-breaking novelty, I think the paper provides useful insights and could be accepted after revisions.

---

### Official Review · Reviewer_dUvT · 2024-07-18
**A great demonstration of deep reinforcement learning (DRL) on a quadrupled system, but lack baseline comparisons and the need of DRL is unclear.**

**Originality:** 2
**Technical Quality:** 3
**Clarity Of Presentation:** 4
**Potential Impact:** 2
**Recommendation:** 2
**Confidence:** 3

**Review:**

### Significance:
- The paper introduces a customized and low-cost quadrupled robot and a nice setup, which would be useful for future research about extraterrestrial exploration if open-sourced


### Clarity:
- The paper is well-written. However, it seems that no appendix was submitted except for a supplementary video

### Originality:
- The paper lacks originality. The approach isn’t new, and limited insights are provided. More specifically, learning policies in simulation with highly engineered rewards with domain randomization and then performing sim-to-real to hardware is commonly used.

### Weakness
- Lack of baseline comparisons: I believe an obvious baseline would use classical controllers such as PIDs + model identification. It is unclear why we need to apply DRL here when we can use classical methods, which potentially will work well, too
- As mentioned earlier, limited new insights are provided. The paper takes the usual approach for sim-to-real and adapts it to a new hardware system
- The reward is highly engineered, and the system requires a MoCap system, which limits the potential for real applications

**Quality Of The Limitations Section:**

3

**Questions For Rebuttal:**

- If possible, please add more comparisons to baselines, especially with classical controllers, to validate the need for using reinforcement learning in this setting
- In Fig. 9 and Fig. 10, I assume the errors are plotted, but it isn’t clear from the description text and the figures’ caption. Also, in Fig.9 and Fig. 10, why is only one trial plotted? Also, the same for Fig.7 and Fig. 8.
- Can the authors clarify why the actuator model is important for successful sim2real transfers? Also, where is the actuator model in Fig. 6?
- Can authors elaborate on the difference between the regulation and velocity tracking problems and the paper was formulated as a regulation problem?

**Robotics Focus:**

4

**Summary Of Paper:**

The paper describes the design of a fast attitude controller for a customized quadrupled robot platform. The attitude controller is learned using PPO via simulated experiences using Isaac Sim. The policy is transferred to real hardware in two tests: free falling and  pole rotating using domain randomization. The learned policy seems to work well on hardware.

**Summary Of Recommendation:**

The paper is a nice technical write-up but lacks novelty to become a science paper as limited new insights are provided.

---

### Author Rebuttal · Authors · 2024-08-11

The attached file contains two documents: The revised paper manuscript and a supplementary document.

In particular, we conduct a baseline comparison with a Model Predictive Control (MPC)-based solution to the attitude control problem, an ablation study of the motor model, an evaluation of the policy’s velocity tracking performance, an evaluation of a policy trained with a reduced reward function, and finally more details on the proposed robotic platform. We also change the original paper to better reflect the contributions of the work and improve the clarity of certain passages. Modifications to the main paper are presented with blue color.

---

### Decision · Program_Chairs · 2024-09-04

**Decision:**

Accept

**Comment:**

# Strengths:
Multiple reviewers acknowledge the writeup clarity, effort in the experiments, and the detailed insights presented in the paper

# Weakness:
Reviewers are failing to recognize the contribution of the proposed work beyond well-known techniques (domain randomization, sim2real, etc) in legged locomotion. Lack of baseline comparisons has also been raised.

# Recommendations:
It is recommended to improve the positioning of the paper. Authors are encouraged to clarify the contributions made by the paper and improve baselines and ablations. Not necessarily, all papers need a novel method. Often well-known techniques applied in interesting ways move the needle in a tough problem. Leverage improvements in baselines and ablations to highlight such contributions and strengthen the submissions.

# Rebuttals
The authors added new experiments and clarified concerns raised by the reviewer. Two authors increased their scores. Reviewers are split on this paper - two in favor and one not in support. Based on the points raised by all reviewers, clarity of presentation, experimental details, and the contributions the paper will make to the field AC recommends acceptance of the paper. Authors are strongly advised to make a best care effort to address all comments from all reviewers, especially dUvT.